

# Managing Soil Nitrogen Surplus: The Role of Winter Cover Crops in N₂O Emissions and Carbon Sequestration

Victoria Nasser[1], René Dechow[2], Mirjam Helfrich[2], Ana Meijide[3], Pauline Sophie Rummel[1,4], Heinz-Josef Koch[5], Reiner Ruser[6], Lisa Essich[6], Klaus Dittert[1]

[1]Department of Crop Sciences, Georg August University of Göttingen, Göttingen, 37075, Germany
[2]Thünen Institute of Climate-Smart Agriculture, Braunschweig, 38116, Germany
[3]Environment Modeling, Institute of Crop Science and Resource Conservation (INRES), University of Bonn, Bonn, 53113, Germany
[4]Department of Biology, Microbiology, Aarhus University, Aarhus, 8000, Denmark
[5]Department of Agronomy, Institute of Sugar Beet Research, Göttingen, 37079, Germany
[6]Department of Fertilization and Soil Matter Dynamics (340i), Institute of Crop Science, University of Hohenheim, Stuttgart, 70599, Germany

*Correspondence to*: Victoria Nasser (victoria.nasser@uni-goetingen.de)

**Abstract.** Cover crops are acclaimed for enhancing the environmental sustainability of agricultural practices by aiding in carbon (C) sequestration and reducing losses of soil mineral nitrogen (SMN) after harvest. Yet, their influence on nitrous oxide (N₂O) emissions—a potent greenhouse gas—presents a complex challenge, with findings varying across different studies. This research aimed to elucidate the effects of various winter cover crops—winter rye (frost-tolerant), saia oat (frost-sensitive grass), and spring vetch (frost-sensitive legume)—against a control of bare fallow on SMN dynamics, N₂O emissions and C sequestration.

While cover crops efficiently lowered SMN levels during their growth, they also increased N₂O emissions in comparison to bare fallow conditions. Notably, winter frost events triggered significant emissions from the frost-sensitive varieties. Moreover, the practices of residue incorporation and soil cultivation were associated with increased N₂O emissions across all cover crop treatments. Winter rye, distinguished by its high biomass production and nitrogen (N) uptake, was linked to the highest cumulative N₂O emissions, highlighting the impact of biomass management and cultivation techniques on N cycling and N₂O emissions.

Cover crop treatment lead to a slight increase in direct N₂O emissions (4.5±3.0, 2.7±1.4, and 3.1±3.8 kg N₂O-N ha$^{-1}$ for rye, oat, and vetch, respectively) compared to the fallow (2.6±1.7 kg N₂O-N ha$^{-1}$) over the entire trial period (16 months). However, the potential of non-legume cover crops to reduce indirect N₂O emissions compared to fallow (0.3±0.4 and 0.2±0.1 kg N₂O-N ha$^{-1}$ a$^{-1}$ for rye and oat respectively) and their contribution to carbon sequestration (120-150 kg C ha$^{-1}$ a$^{-1}$ over a period of 50 years when growing cover crops every fourth year) might partially counterbalance these emissions. Thus, while cover crops offer environmental benefits, their net impact on N₂O emissions necessitates further exploration into optimized cover crop selection and management strategies tailored to specific site conditions to fully leverage their ecological advantages.





## 1 Introduction

The use of cover crops is currently being strongly promoted in many countries due to the multifaceted agro-ecological benefits
they offer. They positively influence soil physical, chemical, and biological properties, enhancing soil water retention and
aiding in weed and disease control through competitive interactions and pest cycle disruptions (Adetunji et al., 2020; Araújo
et al., 2021). The incorporation of cover crop residues boosts soil microbial biomass and activity, thereby enriching biodiversity
and providing habitats for beneficial insects (Elhakeem et al., 2019; Finney et al., 2017). When managed effectively, cover
crops can enhance the yield of subsequent main crops (Adetunji et al., 2020). Grunwald et al. (2022) observed that cover-
cropping prior to sugar beet cultivation improved soil physical properties, facilitating early growth of sugar beet, which is
crucial for high sugar yields (Malnou et al., 2006). However, the effect of cover crop on soil water storage, succeeding crop
yield, and water-use efficiency may not be consistent in all regions (Wang et al., 2021).

Cover crops may also influence nitrous oxide ($N_2O$) emissions, a potent greenhouse gas. The agricultural sector accounts for
about 60% of global anthropogenic $N_2O$ emissions (Masson-Delmotte et al., 2021). These emissions have increased since 1980
due to increased nitrogen (N) fertilizer and manure usage (Davidson, 2009; Tian et al., 2020). Approximately half of the N
applied to agricultural fields is absorbed by crops, while the remainder is subject to loss into the atmosphere as $NH_3$, $NO$, $N_2O$
and $N_2$ or loss to groundwater and surface water primarily in the form of nitrate ($NO_3^-$) (Galloway and Cowling, 2002). In
temperate regions, N losses are exacerbated during periods of high precipitation (Gabriel and Quemada, 2011). Cover crops
can mitigate these losses by absorbing excess soil mineral N (SMN) after harvest (Abdalla et al., 2019), reducing $NO_3^-$ leaching
and runoff, and thereby reducing the need for N fertilization in subsequent crops (Constantin et al., 2011; Hanrahan et al.,
2021; Nouri et al., 2022; Tonitto et al., 2006). Leguminous cover crops further contribute to soil N through atmospheric fixation
(Parr et al., 2011). They may also aid in carbon (C) sequestration when used as green manure, enhancing soil C stocks (Poeplau
and Don, 2015). Nevertheless, studies on the net effects of cover crops on $N_2O$ emissions have yielded mixed results (Basche
et al., 2014; Guenet et al., 2021).

Cover crop residue management and soil cultivation practices play a crucial role in $N_2O$ emission dynamics and magnitudes.
Frost-tolerant crops can be terminated using various methods in preparation for the next cash crop, while frost-sensitive ones
are typically terminated by winter frosts (Storr et al., 2021; Wayman et al., 2015). In some cases, greater soil disturbance in
conventional tillage was found to increase $N_2O$ emissions compared to reduced tillage or no-till (Chatskikh et al., 2008). On
the other hand, ploughing of heavy soils was found to significantly reduce $N_2O$ emissions (Rochette et al., 2008). Soil
incorporation of cover crop residues increases $N_2O$ emissions compared to surface placement, likely due to accelerated
decomposition following increased contact with soil microorganisms (Basche et al., 2014; Lynch et al., 2016).

Crop residues are generally seen as contributors to $N_2O$ emissions because of their N and C content. According to the IPCC
(2019), about 1% of the N in crop residues is converted to $N_2O$. The biochemical properties of cover crops, such as the C:N
ratio, are critical in influencing residue decomposition rates and subsequent $N_2O$ emissions (Lynch et al., 2016). Residues with
lower C:N ratio are decomposed faster, leading to higher $N_2O$ emissions (Basche et al., 2014; Chen et al., 2013; Fosu et al.,



2007). The developmental stage of cover crops at termination is also relevant due to its impact on residue composition and the amount of $N_2O$ emissions after incorporation (Balkcom et al., 2015). Early termination of cover crops, which are typically characterized by lower C:N ratios, results in higher $N_2O$ emissions (Abalos et al., 2022). Legume cover crops generally have lower C:N ratios and result in higher $N_2O$ emissions than non-legumes (Basche et al., 2014; Muhammad et al., 2019).

While crop residue quality plays the biggest role in predicting crop residue-induced $N_2O$ emissions, environmental factors such as soil pH, soil N, available soil organic C (SOC), water-filled pore space (WFPS) and temperature also play a role (Abalos et al., 2022). In temperate cold humid zones, freeze-thaw cycles can lead to significant $N_2O$ emission peaks, contributing substantially to the annual cropland $N_2O$ emissions in these regions (Goodroad and Keeney, 1984; Lemke et al., 1998; Wagner-Riddle et al., 2017).

The majority of studies investigating cover crops in agricultural rotations have focused on short-term $N_2O$ emissions, during either cover crop growth or following residue incorporation, leaving a knowledge gap about year-round $N_2O$ emissions in systems including cover crop cultivation (Muhammad et al., 2019).

In numerous meta-studies it has been shown that cover crops increase SOC stocks (Abdalla et al., 2019; Blanco-Canqui, 2022; Blanco-Canqui et al., 2015; Bolinder et al., 2020; Poeplau and Don, 2015). Thus, soil C sequestration from cover crops needs
to be considered when assessing the effect of cover crops on GHG emissions from croplands. The magnitude of GHG savings depend on site-specific conditions and additional C inputs from cover crops. Soil organic models have been applied to model effects of cover crops on SOC sequestration (Poeplau and Don, 2015; Seitz et al., 2023). In these model applications, the modelled effects are sensitive to the additional C inputs from cover crops that vary between cover crop species.

The objectives of this study were to evaluate the influence of different cover crop species, varying in their frost tolerance and
biochemical composition, on both short-term and long-term $N_2O$ emissions, as well as on the dynamics of SMN in a cover crop – sugar beet – winter wheat rotation on Luvisol soil type frequently used for sugar beet cropping in Germany. Additionally, we aimed to investigate the drivers of $N_2O$ emissions, such as soil temperature, soil moisture, SMN concentrations, and the quantity and composition of the biomass of the incorporated cover crops. Our study seeks to balance the advantages and disadvantages of using cover crops concerning their impact on $N_2O$ emissions, SMN dynamics and SOC
sequestration.

We hypothesize that:

i) Non-legume cover crops take up excess N in autumn, leading to decreased SMN concentrations in winter and consequently reducing $N_2O$ emissions compared to fallow.

ii) The freezing of frost-sensitive cover crops is followed by an increase in SMN values, resulting in higher $N_2O$ emissions
during winter.

iii) Incorporating cover crops with larger biomass residues results in higher $N_2O$ emissions in the subsequent main crop.

iv) Due to their faster decomposition, cover crops with lower C:N ratios cause higher $N_2O$ emissions after incorporation compared to cover crops with higher C:N ratio.





v) Carbon inputs from aboveground and belowground biomass, which contribute to soil organic C stocks, vary significantly
between cover crop species.

## 2 Materials and methods

### 2.1 Study sites and experimental design

Field trials were conducted in central (Göttingen) and southern Germany (Ihinger Hof, Hohenheim), with replicated fields at each site, established in two consecutive years systematically named to reflect their location and establishment year.
Specifically, the trials initiated in 2018 were labelled G18 for Göttingen and H18 for Hohenheim, while those initiated in 2019 were labelled G19 and H19, respectively. Each of these trials began in autumn and continued for approximately 18 months, with the 2018 trials ending in March 2020 and the 2019 trials in March 2021. Different fields were used at each site to avoid residual effects between trials. The elevations were 160 m asl for G18, 150 m asl for G19, and 480 m asl for both H18 and H19. The soils at both sites were classified as Luvisols (IUSS, 2015), with topsoil organic matter content of 20–30 g kg⁻¹ and
pH 7.0–7.5 (in 0.0125 $M$ CaCl$_2$). Soils in Göttingen had 12-14% clay 3-15% sand, while in Hohenheim it had 26-28% clay and about 2-3% sand.

Long-term climate data (1991-2020) from the German Meteorological Service (DWD, 2023) recorded average annual precipitation of 624 mm and temperature of 9.4°C for Göttingen, and 701 mm precipitation with a temperature of 9.1°C for Hohenheim. Meteorological data during the study were collected from stations located at the field sites.

Field pea (*Pisum sativum* L.) was cultivated prior to the experiments due to their high residual soil NO$_3^-$ content and potential for NO$_3^-$ leaching (Voisin et al., 2002). Pea straw was left on the fields and incorporated into the soil by ploughing or deep rigid tine cultivator tillage. A randomized complete block design with four replications was set up at each site and year. Plot sizes were 21x17 m in Göttingen and 30x19 m in Hohenheim, with sampling restricted to subplots of 2.7x14 m and 3x12 m, respectively. Three different cover crops were sown in autumn: saia oat (*Avena strigosa* Schreb. var. "Pratex") and winter rye
(*Secale cereale* L. var. "Traktor"), representing frost-sensitive and frost-tolerant grasses, respectively, and spring vetch (*Vicia sativa* L. var. "Mirabella"), a frost-sensitive leguminous cover crop. These were compared to a bare fallow during the cover crop cultivation period. Seedbed preparation was done using a disc or rotary harrow. Sowing was done on 29/08/2018 in G18, 12/09/2018 in H18, 8/8/2019 in G19 and 4/09/2019 in H19 at rates of 120 kg ha⁻¹ for rye, 80 kg ha⁻¹ for oat, and 90 kg ha⁻¹ for vetch, with row spacing of 12.5 cm in Göttingen and 15.0 cm in Hohenheim. During the cover crop phase, herbicides were
applied in autumn for weed control in fallow plots and cover crop plots were not fertilized. In March of year following establishment, cover crops were treated with glyphosate to terminate any plant growth after winter. To safeguard an optimal seedbed for sugar beet seedlings, rye was ploughed to a depth of 30 cm due to its extensive crown root and stem base material, while other treatments were tilled to 15 cm using a short disc harrow or tine cultivator.

Sugar beet (*Beta vulgaris* L. var. "Lisanna") followed as the first main crop, sown at 45-50 cm row spacing and 90,000–95,000
plants ha⁻¹ density. Nitrogen fertilization for sugar beet followed the German fertilizer ordinance amounting to a total N



demand of 180 kg N ha⁻¹ depending on SMN in March, leading to 60 kg N ha⁻¹ for G18, 100 kg N ha⁻¹ for G19, 130 kg N ha⁻¹ for H18, and 140 kg N ha⁻¹ for H19. In the G19 trial, sugar beet following the rye treatment inadvertently received by mistake 200 kg N ha⁻¹, double the intended amount of 100 kg N ha⁻¹, which was applied to the other treatments. After harvest, sugar beet leaves were incorporated by 12-15 cm deep cultivator tillage before sowing winter wheat (*Triticum aestivum* var.
"Nordkap") in October. Trials ended with the first N fertilization of winter wheat next March.

**2.2 Plant and soil sampling, analyses and calculations**

The aboveground biomass of cover crops was assessed by the end of November at each site-year using four sampling points within each plot, each covering an area of 0.5 m². To determine the dry matter (DM) content of the cover crop biomass, a subsample was mashed, and then dried at 60 °C for 48 hours. The resulting dry weight was then used to calculate the DM
biomass. Elemental analysis was performed on the plant material to determine C and N concentrations, which were subsequently utilized to calculate the C and N contents of the aboveground cover crop biomass. Composite soil samples of five subsamples were taken biweekly from the topsoil (0-30 cm depth) using a 30 mm diameter auger. Samples were stored at -20°C until analysis. To determine SMN content (i.e. the sum of $NO_3^-$ and $NH_4^+$), soil samples were extracted with 0.0125 $M$ $CaCl_2$ solution in the ratio 1:4 (w:v) and analyzed according to VDLUFA A 6.1.4.1 (VDLUFA, 1991). Gravimetric water
content was measured on subsamples dried at 105°C for 24 h. To determine soil bulk density, six undisturbed soil cores (250 cm³) per plot were taken in winter from the topsoil (3-8 cm, 13-18 cm and 23-28 cm), dried at 105°C for 24 h before weighing. Soil bulk density was calculated according to Eq. (1):

$$\rho b = \frac{Ms}{Vt},$$    (1)

Where $\rho b$ is the bulk density (g/cm³), $Ms$ is dry soil weight (g) and $Vt$ is Soil volume (cm³).
Water-filled pore space was calculated from the gravimetric water content and the bulk density according to Eq. 2:

$$WFPS = \frac{\theta g * \rho b}{1 - \frac{\rho b}{\rho s}} * 100\%,$$    (2)

Where WFPS is the water-filled pore space (%), $\theta g$ is the gravimetric water content (g/g⁻¹), $\rho b$ is the bulk density (g/cm³), $\rho s$ is the assumed particle density (2.65 g cm⁻³).

**2.3 Gas sampling and analysis**

The closed chamber method was used for the $N_2O$ flux measurements (Hutchinson and Mosier, 1981). Samples were taken weekly or more frequently after fertilization, high precipitation, or frost-thaw events. Two types of chambers were used at the study site in Göttingen: round chambers (60 cm diameter, 45 cm height) for cover crop and winter wheat phases, and rectangular chambers (72 cm length, 27 cm width, 18 cm height) for sugar beet phase. Chambers were made of white opaque PVC and sealed with rubber straps or brackets. At the study site Hohenheim fluxes were measured using dark, vented chambers
with an inner diameter of 0.3 m as described in detail by Flessa et al., (1995). Gas samples were collected through a septum



using a 30 mL syringe and stored in pre-evacuated 12 mL vials (Exetainer, Labco Limited, UK). At both study sites, laboratory analysis employed a SCION 456-GC gas chromatograph with an electron capture detector (ECD) for $N_2O$ and a thermal connectivity detector (TCD) for $CO_2$. Samples were introduced using a Gilson auto sampler (Gilson Inc., Middleton, WI, USA). Data processing was performed using '*CompassCDS*' software. The analytical precision of the gas chromatograph was

determined monthly by repeated measurements of certified standard gases (307, 760, and 6110 ppb $N_2O$. 201, 550 and 2500 ppm $CO_2$) and the coefficient of variation was consistently < 2%. Mass concentrations were calculated from molar concentrations using the ideal gas equation considering the chamber temperature.

Flux rates were calculated using the "*gasfluxes*" R package (Fuss and Hueppi, 2020), selecting models based on the Akaike information criterion and Kappa value. Cumulative $N_2O$ emissions for the different cropping phases were estimated using

"*aggfluxes*" function with linear interpolation between measurement dates and summed to result in the cumulative fluxes of the entire trials. Potentially mitigated indirect $N_2O$ emissions were estimated by multiplying the late autumn N uptake of cover crop shoots by the (IPCC, 2019) factors: the $N_2O$ emission factor for indirect emissions due to N leaching and runoff ($EF_5$ = 0.011 kg $N_2O$ –N per kg N) and the factor for N losses by leaching and runoff in wet climates ($Frac_{LEACH}$ = 0.24 kg N per kg N). $CO_2$ equivalents ($CO_{2eq}$) were calculated by using the $N_2O$ global warming potential of 273 (IPCC, 2022).

**2.4 Data and statistical analyses**

Data processing and analyses were carried out in R version 4.2.2 (R Core Team, 2023). For all statistical analyses, the significance level was set to $p < 0.05$. $N_2O$ fluxes and cumulative emissions were $log_{10}$ transformed to ensure normal distribution of data and residue variance homogeneity. Variance homogeneity and approximate normality of residuals was assessed using diagnostic plots. Flux rates that were strongly negative (i.e. lower than -60 g $N_2O$-N $ha^{-1}$ $d^{-1}$) as well as those with standard

errors larger than 120 µg $N_2O$-N $m^{-2}$ $h^{-1}$, indicating high uncertainty, were excluded. In addressing the discrepancy in fertilizer application in the G19 trial, data from rye treatment, which had received the double fertilizer dose, were included in analyses, plots and tables specific to individual site-year evaluations. However, for comprehensive analyses that combined data across all site-years, the rye treatment data from G19 during the sugar beet and winter wheat phases were excluded to maintain consistency and comparability across the study. ANOVA was performed on linear models for treatment differences, with

Tukey HSD tests for post-hoc comparisons. A generalized least-squares regression model assessed cover crop impact on cumulative $N_2O$ emissions, average SMN and WFPS across site-years, using the '*nlme*' package (Pinheiro et al., 2023). Variance heterogeneity was addressed using the variance structure $\sigma^2 \times |y|^{2\delta}$ (Zuur et al., 2009), applied when found significant. Upon identifying a significant cover crop treatment effect via ANOVA, pairwise mean comparisons, with Tukey-adjusted p-values, were performed on the estimated marginal means through the '*emmeans*' package (Lenth et al., 2023). Impact of

environmental variables on $N_2O$ flux was analyzed using linear mixed-effects models with '*lmer*' function from '*lme4*' package (Bates et al., 2015,), with predictors standardized using Z-score normalization from '*scales*' package (Wickham et al., 2023) for comparability and to address scale differences. The models included standardized SMN, WFPS, soil temperature, and cover crop treatments as fixed effects, and site-year as a random effect to account for variability across different sites and years.





**2.5 Modelling changes in soil organic C stocks**

The effect of cover crops on SOC sequestration was simulated over a period of 50 years for two different crop rotations, common in German agricultural practice. Crop rotation CR1, with a cover crop embedded every second year, had the sequence cover crop/bare fallow – sugar beet – winter wheat – cover crop/bare fallow – silage maize – winter wheat with an application of 30 m$^3$ digestate from biogas plants before seeding of maize. Crop rotation CR2 had the crop sequence cover crop/bare fallow – sugar beet - winter wheat – winter rape – winter wheat, with a cover crop in every 4$^{th}$ year. No organic fertilizer was applied

in crop rotation CR2. In the scenarios, it was assumed that the above-ground crop residues for winter wheat, sugar beet and winter oilseed rape remain in the field. Scenarios were done for every treatment and effects of cover crops have been quantified by subtracting modelled C stocks of the control (bare fallow instead of cover crop) from modelled C stocks of cover crop treatments. Because we use simple first order kinetic models, modelled SOC change between scenarios is linear dependent on differences in initial C stocks and C inputs between control and cover crop scenarios, meaning that in our setup frequency and

biomass production of grown cover crops mainly control modelled effects on SOC stocks.

    The long-term potential for changes in SOC content due to the cultivation of cover crops was estimated using a model ensemble (Seitz et al., 2023) consisting of the RothC (Coleman and Jenkinson, 1996) and C-Tool (Taghizadeh-Toosi et al., 2014) models implemented in R using the SoilR package (Sierra et al., 2012) in combination with three allometric functions for calculating the C input from shoot and root residues of main crops in dependence on yields (Franko et al., 2011; Jacobs et al., 2020;

Rösemann et al., 2021). If no yield information was available for main crops, documented yields from German agricultural soil inventory sites (Jacobs et al., 2020) within a radius of 50 km were used and averaged. In order to integrate the intercropping effects on the biomass of the subsequent crop into the modelling, variant-specific yields of the first and second subsequent crop were used in the calculation of C inputs from crop root residues of the main crop.

    The C inputs of the cover crops were taken from measured above-ground biomass (experimental data) and root-shoot ratios,

which in turn were derived on the basis of measurements of above- and below-ground biomass of the cover crops grown at the Göttingen site. In order to derive the C input for root biomass from cover crops for the horizon of interest of 0 - 30 cm profile depth, an approach according to Gale and Grigal (1987) was used to describe the root distribution as a function of depth:

$$Y = 1 - \beta^d \tag{3}$$

    Here, Y is the root fraction increasing with depth, β is a parameter and d is the depth of the profile in cm. Plant-specific

parameters β were determined via calibration on the basis of plant-specific available data on below-ground biomass at the Göttingen site (Figure S1, Supplementary Table S5). Estimated proportion of roots in the 0-30 cm depth profile is higher for winter rye (89 %) than for saia oat and spring vetch (79 % each). In relation to the profile depth of 0-30 cm considered here, this results in root-to-shoot ratios of 0.24, 0.13 and 0.33 for saia oat, spring vetch and winter rye, respectively. It was assumed that C input from root exudates corresponds to 31% of the input from root C according to Jacobs et al. (2020).





Weather data in monthly resolution was taken from DWD grid data on monthly precipitation, temperatures and global radiation for the years 2018-2021 (DWD, 2023). These time series were repeated to get weather data time series with a duration corresponding to the simulation period of 50 years.

# 3 Results

## 3.1 Weather conditions and soil WFPS

The experimental trials experienced weather conditions that were marginally warmer and drier than the long-term averages for both sites (Table 1). The year 2018 was exceptionally dry and warm in Germany, with recorded rainfall of only 430 mm in Göttingen and 526 mm in Hohenheim. Air temperatures typically ranged between -5 and 30 °C, with higher readings during summer and a notably cold week in mid-February 2021, where temperatures dropped to -23 °C in Göttingen and -17 °C in Hohenheim. During this period, a thick layer of snow insulated the soil, preventing freezing. Soil temperatures mostly

fluctuated between 0-25 °C. However, several soil-frost events were recorded, in which soil temperatures at 5 cm depth dropped below 0 °C. In Göttingen, a weeklong soil-frost event at the end of January 2019 saw soil temperatures fall to -4.6°C, and a one-day event occurred in early January 2020. Hohenheim experienced a milder frost event at the end of January 2019, reaching -1.5°C, and a two-day event in mid-January 2021, with temperatures down to -1.9°C. Göttingen generally had slightly warmer conditions with higher precipitation than Hohenheim throughout most cropping phases (Table 1).

Water-filled pore space exhibited a consistent trend across all site-years, starting low in autumn and gradually increasing through autumn and winter, peaking around February. Levels then declined during spring and summer, reaching their lowest in August. Heavy summer rainfall occasionally caused short-term spikes in WFPS (Figure 1). Variations in WFPS were observed between site-years and cropping phases, although these differences were not consistent throughout the entire duration of the study (Table 1). Average WFPS values in the G18 trial, under cover crops were marginally lower compared to the fallow

treatment during the cover-cropping phase, yet this difference was not statistically significant (Supplementary Table S1). Across all treatments and phases, including the cover-cropping phase and the ones that followed, no significant differences in WFPS were identified between treatments (Table S1).



**Table 1.** Weather conditions and soil water filled pore space (WFPS) across different cropping phases and site-years. Means ± (SD). Different lowercase letters indicate statistically significant differences between site-years within each column (p<0.05).

| | Cover crop phase | Sugar beet phase | Winter wheat phase | Entire trial |
|---|---|---|---|---|
| Mean temperature [°C] | | | | |
| G18 | 7.1 (5.6) | 16.1 (5.2) | 6.1 (4.2) | 9.6 (6.7) |
| H18 | 5.4 (5.4) | 14.8 (4.8) | 3.8 (3.5) | 8.3 (6.8) |
| G19 | 8.1 (6) | 15.6 (4.3) | 4.1 (5.8) | 9.5 (7.1) |
| H19 | 6.2 (5) | 15.7 (4.1) | 3.9 (5.1) | 8.4 (6.8) |
| Mean weekly precipitation [mm week$^{-1}$] | | | | |
| G18 | 13.3 (14.7) | 11 (12.4) | 16.6 (12) | 13.6 (13.3) |
| H18 | 9.1 (9.8) | 13.9 (14.4) | 10.5 (8.5) | 11.2 (11.5) |
| G19 | 17.7 (18.9) | 16.4 (21.1) | 7.5 (6.9) | 14.6 (17.8) |
| H19 | 9.7 (8.1) | 11 (12.7) | 6.9 (7.7) | 9.3 (9.6) |
| Soil moisture [%WFPS] | | | | |
| G18 | 57 (24) a | 54 (20) a | 79 (16) a | 62 (24) a |
| H18 | 66 (23) b | 47 (14) b | 71 (9) b | 62 (21) a |
| G19 | 65 (20) b | 47 (14) b | 70 (11) bc | 61 (19) a |
| H19 | 75 (12) c | 54 (13) a | 69 (12) c | 67 (15) b |

## 3.2 Winter survival and biomass characteristics of cover crops

In the trials established in 2018 (G18 and H18), the frost event in January 2019 terminated the frost-sensitive oat and vetch in G18, as well as oat in H18, while the vetch in H18 was only partially terminated by the frost. The following year, the trials experienced milder conditions, affecting only vetch in G19 in late November 2019, while oat reached early senescence due to a virus infection. However, oat and vetch in H19, along with rye in all site-years, survived the winter, necessitating termination in March using glyphosate.

By late autumn, significant differences in the aboveground biomass DM were observed across all site-years, except for H18. Rye consistently had higher aboveground biomass DM (3.6±0.8 t ha$^{-1}$) compared to oat and vetch (2.7±0.8 and 2.2±0.7 t ha$^{-1}$, respectively). The C content in cover crop aboveground biomass was highest in rye (1.2±0.2 t ha$^{-1}$), followed by oat (0.9±0.2 t ha$^{-1}$), and lowest in vetch (0.7±0.2 t ha$^{-1}$). Rye also had the highest N uptake (103±37 kg N ha$^{-1}$), while oat and vetch showed variable uptakes (75±23 and 76±21 kg N ha$^{-1}$, respectively). A strong correlation was observed between cover crop DM formation and both C content and N uptake (r>0.9). The C:N ratio of cover crop shoot biomass was consistently lower for vetch (<10) compared to rye and oat (Table 2).



**Table 2.** Dry matter biomass (DM), carbon (C) and nitrogen (N) contents, and C:N ratio of winter cover crops in late autumn across site-years. Means (n=4 for individual trials, n=16 for averages across all site-years) ±(SD). Significant differences between cover crop species within each category are denoted by different lowercase letters (p<0.05).

|  | G18 | H18 | G19 | H19 | Mean |
|---|---|---|---|---|---|
| DM (t ha$^{-1}$) |  |  |  |  |  |
| Winter rye | 4.4 (0.2) a | 2.7 (0.4) | 3.5 (0.2) a | 4.3 (0.5) a | 3.6 (0.8) a |
| Saia oat | 3.8 (0.2) b | 2.6 (0.6) | 2.1 (0.3) b | 2.3 (0.3) b | 2.7 (0.8) a |
| Spring vetch | 1.9 (0) c | 2.7 (1.9) | 2.4 (0.3) b | 1.9 (0.6) b | 2.2 (0.7) b |
| C content (t C ha$^{-1}$) |  |  |  |  |  |
| Winter rye | 1.2 (0.1) a | 1.1 (0.1) a | 1 (0.1) a | 1.7 (0.2) a | 1.2 (0.2) a |
| Saia oat | 1.1 (0.1) a | 0.9 (0.1) a | 0.7 (0.1) b | 1 (0.1) b | 0.9 (0.2) b |
| Spring vetch | 0.7 (0) b | 0.4 (0) b | 0.9 (0.1) ab | 0.7 (0.2) b | 0.7 (0.2) c |
| N content (kg N ha$^{-1}$) |  |  |  |  |  |
| Winter rye | 109 (9) a | 64 (12) | 99 (2) a | 172 (9) a | 103 (37) a |
| Saia oat | 104 (5) a | 61 (11) | 50 (4) b | 82 (10) b | 75 (23) b |
| Spring vetch | 77 (1) b | 41 (3) | 92 (11) a | 78 (22) b | 76 (21) ab |
| C:N ratio (-) |  |  |  |  |  |
| Winter rye | 11 (0.7) a | 17.1 (0.9) a | 10.4 (0.5) a | 9.6 (0.6) a | 12.5 (3.2) a |
| Saia oat | 10.9 (0.6) a | 15.7 (0.9) a | 13.6 (0.8) b | 11.7 (0.5) b | 12.8 (1.9) a |
| Spring vetch | 8.6 (0.4) b | 8.8 (0.4) b | 9.6 (0.4) a | 8.4 (0.1) a | 8.9 (0.6) b |

## 3.3 Soil mineral N dynamics

Topsoil mineral N was predominantly composed of $NO_3^-$, with ammonium ($NH_4^+$) levels remaining below 10 kg N ha$^{-1}$ on most sampling dates. At the onset of trials, SMN levels varied among sites and years, with the highest average in G18 (80±20 kg N ha$^{-1}$), followed by G19 (48±14 kg N ha$^{-1}$) and the lowest in Hohenheim (29±8 and 33±9 kg N ha$^{-1}$ for H18 and H19, respectively). No significant differences in initial SMN levels were found between treatments within each trial (Figure 1, Table S2).

A brief increase in SMN was observed in the first 2-4 weeks following pea straw incorporation, soil cultivation, and cover crop sowing. During autumn, SMN levels gradually decreased in cover crop treatments, while remaining elevated in bare fallow. By the end of November, significantly lower SMN levels were recorded in all cover crop treatments compared to fallow across all site-years (Figure 1, Table S2). The lowest levels were reached around January, with averages of 27±12 kg N ha$^{-1}$ in Göttingen and 8±6 kg N ha$^{-1}$ in Hohenheim.





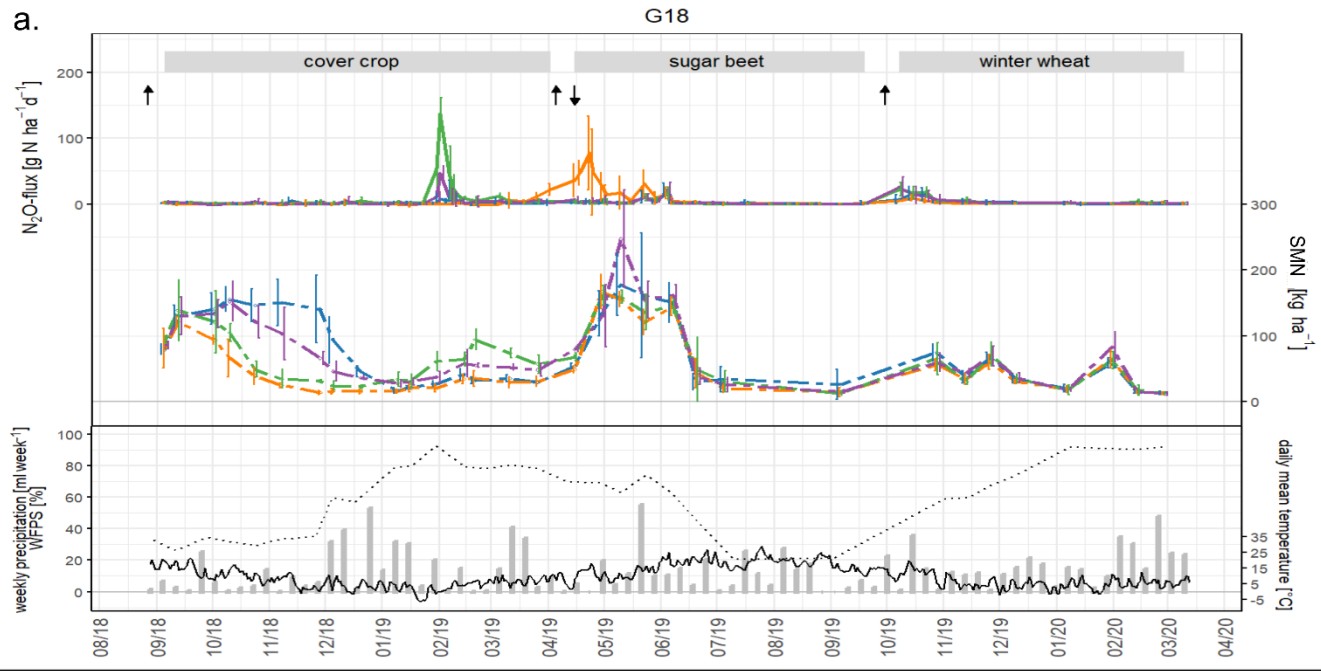

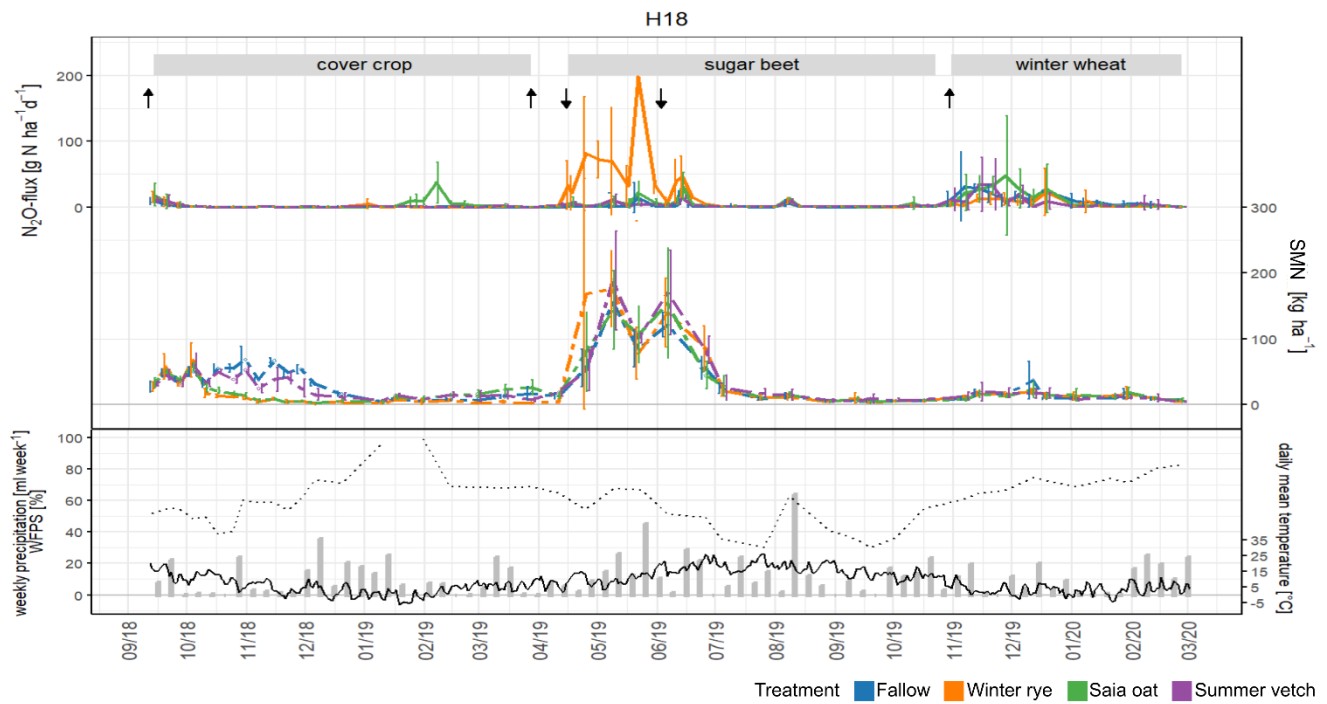





**Figure 1.** Dynamics of N₂O flux rates and topsoil mineral nitrogen (SMN) contents, presented as mean ± SD (n=4). Seasonal changes in topsoil water-filled pore space (WFPS%, dotted black line), daily mean air temperature (continuous black line) and weekly precipitation (gray bars) for Göttingen (upper plot) and Hohenheim (lower plot) throughout the different cropping phases for various cover crop treatments in (a) 2018 trials and (b) 2019 trials. Upward arrows mark soil cultivation events, while downward arrows signify N fertilization of sugar beets.





Following the frost event in late January 2019, a notable increase in SMN was observed in the frost-sensitive oat and vetch
treatments in G18, as well as in oat in H18 (Figure 1a). At the end of the cover crop phase in April, SMN levels were lowest
in rye, with some variation in other treatments across site-years (Figure 1, Table S2). The average SMN values during the
cover-cropping phase were lowest for non-legume cover crops (rye and oat), while bare fallow consistently had the highest
average SMN across all site-years (Figure 2a). Rye exhibited significantly lower average SMN levels than bare fallow in all

site-years (Supplementary Table S3).

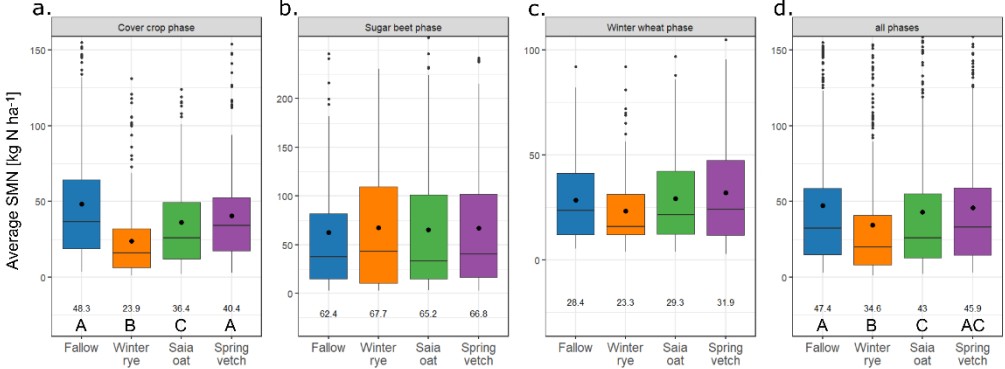

**Figure 2.** Mean topsoil mineral nitrogen (SMN) during the different cropping phases. Mean SMN ± SD (n=12 for rye treatment in the sugar
beet, winter wheat and all phases together, n=16 for all other treatments). Horizontal lines represent the median, large dots and numbers at
the bottom show the mean. Uppercase letters indicate significant differences of Z-standardized Nmin values between the treatments (p<0.05).

Following the incorporation of cover crop residue through soil cultivation, the sowing of sugar beets, and the application of N
fertilizer in April, SMN levels increased to a range of 111-247 kg N ha$^{-1}$ at the onset of the sugar beet phase. The impact of N
fertilization on SMN content was so substantial that any differences between cover crop treatments were not statistically
significant.

As the sugar beet plants increased their N uptake, SMN levels significantly decreased between June and July, reaching their

lowest levels for this phase around September (Figure 1, Table S2). Further N fertilization applied to the rye treatment in June
for the G19 trial, and to all treatments in the H18 trial, led to elevated SMN levels in the corresponding plots. Average SMN
levels for the sugar beet phase were highest in G18, followed by H19 and G19, and were lowest in H18, with no significant
differences between cover crop treatments (Table S3).

After harvest of sugar beet, the following soil cultivation, leaf residue incorporation, and winter wheat sowing, an increase in

SMN levels was observed in all site-years for most treatments (Figure 1). Average SMN levels during the winter wheat phase
were higher in Göttingen than in Hohenheim, with no residual cover crop effect evident, as no significant differences between
treatments were observed in this phase (Table S3).

Over the entire trial duration, the average SMN levels were highest in G18, intermediate in G19, and lowest in both H18 and
H19. The non-legume cover crop treatments resulted in significantly lower SMN values compared to bare fallow across the

entire experimental period, with rye treatment showing the lowest average SMN (Figure 2, Table S3).



**3.4 N₂O flux rates and cumulative emissions**

We observed spatial and temporal variations in $N_2O$ flux rates, but patterns were similar across all site-years. Heavy rainfall and increases in WFPS frequently led to $N_2O$ emission peaks throughout the study (Figure 1). A notable peak in $N_2O$ emissions followed the frost event at the end of January 2019 in the frost-sensitive oat treatment in G18 and H18. The peak emissions

from the oat treatment were significantly higher than in all other treatments and reached 137 ($\pm$24) g $N_2O$-N ha$^{-1}$ d$^{-1}$ in G18, with a smaller peak in H18. A smaller but significant peak was also measured in the vetch treatment in G18 but not H18 (Figure 1a). Compared to other cover crops, oat led to higher cumulative $N_2O$ emissions in three out of four site-years during the cover-cropping phase. However, this difference was only significant for both 2018 trials (G18 and H18). Vetch recorded the highest cumulative $N_2O$ emissions during the cover-cropping phase in G19 (Supplementary Table S4).

Following soil cultivation, sugar beet sowing and N fertilization, significantly higher $N_2O$ emissions were observed in the rye treatment compared to other treatments, lasting for about six weeks from April to June in all site-years (Figure 1). As a result, cumulative $N_2O$ emissions during the sugar beet phase were significantly higher in rye compared to other treatments in three out of four site-years (Table S4).

After the harvest of the sugar beet, the following soil cultivation, incorporation of leaf residue, and sowing of winter wheat,

we observed an increase in $N_2O$ fluxes, with the highest peaks occurring in H19 and the lowest in G18. During the winter wheat phase, the cumulative $N_2O$ emissions were significantly lower in the rye treatment than in the fallow and vetch treatments, which exhibited higher emissions. Cumulative $N_2O$ emissions for the entire trial period were generally highest in H19 and lowest in G18; however, these differences were not statistically significant (Table S4).

**3.5 Synthesis over all site-years**

A distinct pattern emerged when data from all sites and years were combined and analyzed with regard to SMN content during the cover-cropping phase (Figure 2, Table S3). It is noteworthy that non-legume cover crops (specifically, rye and oat) consistently yielded significantly lower SMN levels (24$\pm$24 kg N ha$^{-1}$ for rye and 36$\pm$37 kg N ha$^{-1}$ for oat) compared to the bare fallow (48$\pm$41 kg N ha$^{-1}$) and the legume cover crop (vetch at 40$\pm$32 kg N ha$^{-1}$, p-value < 0.01, n=16). This pattern was particularly pronounced in the autumn, with SMN levels in rye and oat plots declining to 14 $\pm$ 13 kg N ha$^{-1}$ and 20 $\pm$ 16 kg N

ha$^{-1}$, respectively, by the end of November. The vetch treatment exhibited intermediate SMN levels (37$\pm$24 kg N ha$^{-1}$), while the bare fallow maintained significantly higher levels (83$\pm$53 kg N ha$^{-1}$, p-value < 0.001, n=16). Conversely, the cumulative emissions of $N_2O$ during the period of cover cropping were higher in the treatments involving cover crops, with the frost-sensitive oat treatment exhibiting a significant increase compared to the bare fallow (Figure 3, p-value < 0.05, n=16).



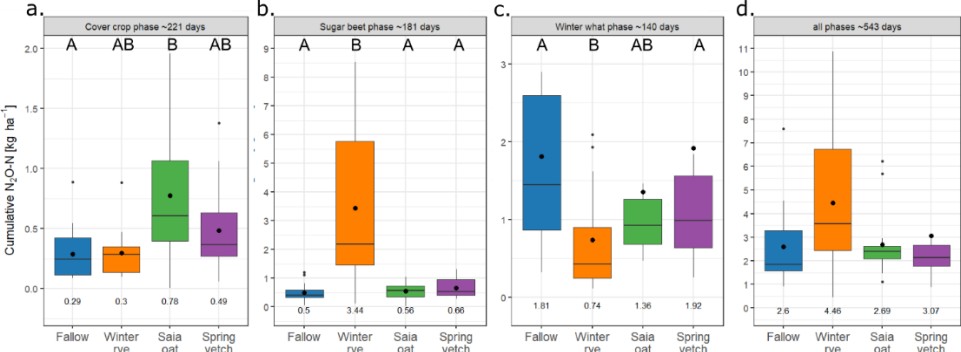

**Figure 3.** Cumulative N$_2$O emissions across the different cropping phases. Mean ± SD (n=12 for rye in sugar beet and winter wheat phases and all phases combined; n=16 for all other treatments). Horizontal lines represent the median, large dots and numbers at the bottom show the mean. Uppercase letters indicate significant differences of Z-standardized cumulative N$_2$O emission values between the treatments (p<0.05).

While all treatments had and maintained similar levels of SMN following fertilization in the sugar beet phase, rye treatment distinctly showed higher cumulative N$_2$O emissions (3.4±2.81 kg N$_2$O-N ha$^{-1}$) compared to other treatments (0.5±0.31, 0.56±0.3 and 0.66±0.34 Kg N$_2$O-N ha$^{-1}$ for fallow, oat and vetch respectively). Interestingly, rye exhibited lower cumulative N$_2$O emissions compared to other treatments in the winter wheat phase, although no residual effect of the cover crops on SMN levels was observed.

In conclusion, throughout the entirety of the trial period, non-legume cover crops resulted in diminished SMN values in comparison to bare fallow, with rye exhibiting the lowest levels on a consistent basis. In contrast, rye resulted in the highest total cumulative N$_2$O emissions, followed by oat and vetch, with fallow having the lowest emissions. Nevertheless, these discrepancies were not statistically significant over the entire trial period.

In analyzing the impact of soil characteristics and cover crop treatments on N$_2$O flux rates, significant relationships were found. Soil temperature as well as WPFS affected N$_2$O flux rates during all cropping phases (p-value<0.001, Table 3). Soil mineral N had a significant effect on N$_2$O fluxes only in the winter wheat phase or when combining all phases together (p-value<0.001). Frost-sensitive cover crop treatments oat and vetch increase N$_2$O flux rates during the cover-cropping and sugar beet phases (p-value<0.001). Frost-tolerant rye cover crop treatment followed by deep ploughing, strongly increased N$_2$O flux rates during the sugar beet phase while reducing them during the following winter wheat phase (p-value<0.001) (Table 3).

**Table 3.** Regression coefficients from a linear mixed-effects model assessing the relationship between N$_2$O flux rates and soil characteristics, along with the impact of different cover crop treatments during various cropping phases and across the entire trial. Significant effects are highlighted in bold. Significance levels are denoted by asterisks (*p<0.05, **p<0.01, ***p<0.001). Standard errors are provided in parentheses. The model includes effects of cover crop treatments (rye, oat, vetch), soil Nmin, WFPS, and soil temperature as fixed effects, along with a random effect for site-year. R-squared values for conditional fits (R$^2$c) and standard deviations (SD) for intercepts across site-years and observations are also listed. The number of observations included in each phase and for the entire trial are specified.





| | Cover crop phase | Sugar beet phase | Winter wheat phase | Entire trial |
|---|---|---|---|---|
| Cover crop winter rye | 0.023 (0.05) | **0.597 (0.043)*** | **-0.299 (0.058)*** | **0.174 (0.033)*** |
| Cover crop saia oat | **0.326 (0.047)*** | **0.193 (0.043)*** | -0.062 (0.057) | **0.174 (0.033)*** |
| Cover crop spring vetch | **0.154 (0.047)*** | **0.208 (0.042)*** | -0.105 (0.058) | **0.095 (0.032)** |
| Topsoil mineral nitrogen (SMN) | 0.032 (0.031) | 0.022 (0.014) | **0.275 (0.061)*** | **0.128 (0.013)*** |
| Water-filled pore space (WFPS) | **0.144 (0.022)*** | **0.508 (0.025)*** | **-0.34 (0.039)*** | **0.109 (0.016)*** |
| Soil temperature | **0.194 (0.023)*** | **0.101 (0.029)*** | **0.165 (0.048)*** | **0.077 (0.016)*** |
| (Intercept) | **0.53 (0.137)*** | **0.811 (0.071)*** | **1.576 (0.109)*** | **0.752 (0.06)*** |
| $R^2c$ | 0.229 | 0.38 | 0.288 | 0.079 |
| Number of Observations | 1365 | 1546 | 1014 | 3925 |

## 3.6 Potentially mitigated indirect N₂O emissions

The mitigation potential for indirect $N_2O$ emissions induced by $NO_3^-$ leaching, calculated from cover crop N uptake in late autumn, was notably higher in rye, averaging $0.27\pm0.1$ kg $N_2O$-N ha⁻¹ a⁻¹, equivalent to approximately $116\pm42$ kg $CO_{2\text{-eq}}$ ha⁻¹ a⁻¹ (Table 4). Oat exhibited a lower mitigation potential, reducing on average $0.2\pm0.0.1$ kg $N_2O$-N ha⁻¹ a⁻¹, or $85\pm26$ kg $CO_{2\text{-eq}}$ ha⁻¹ a⁻¹ (Table 4). The calculation excluded vetch due to the uncertainty in distinguishing the proportion of N uptake from soil versus N biologically fixed.

**Table 4.** Mitigation potential of non-legume cover crops in reducing indirect $N_2O$ emissions induced by N leaching derived from pre-winter N uptake of cover crops. Mean values (n=4 for individual trials, n=16 for overall averages) ± (SD). Statistically significant differences between rye and oat within each trial and overall averages are denoted by different lowercase letters.

| | | Mitigation potential | |
|---|---|---|---|
| | | kg $N_2O$-N ha⁻¹ a⁻¹ | kg $CO_{2\text{-eq}}$ ha⁻¹ a⁻¹ |
| G18 | Winter rye | 0.29 (0.02) | 123 (10) |
| | Saia oat | 0.28 (0.01) | 118 (6) |
| H18 | Winter rye | 0.17 (0.03) | 73 (14) |
| | Saia oat | 0.16 (0.03) | 69 (12) |
| G19 | Winter rye | 0.26 (0.01) a | 112 (3) a |
| | Saia oat | 0.13 (0.01) b | 57 (5) b |
| H19 | Winter rye | 0.45 (0.02) a | 194 (10) a |
| | Saia oat | 0.22 (0.03) b | 93 (11) b |
| Mean | Winter rye | 0.27 (0.1) a | 116 (42) a |
| | Saia oat | 0.2 (0.06) b | 85 (26) b |

## 3.7 Changes in soil organic C

At the Göttingen sites, averaged C inputs from main crops in the control scenarios (no cover crop) were about 4.15 and 4.51 t C ha⁻¹ a⁻¹ for CR1 and CR2, respectively, while averaged C inputs at the Hohenheim site were about 10% lower with 3.76 t C ha⁻¹ a⁻¹ and 4.15 t C ha⁻¹ a⁻¹ (Supplementary Table S6). Crop rotation CR1 leads to lower C inputs than crop rotation CR2, which is mainly due to higher share of winter crops maintaining high amounts of incorporated crop residues, despite cover crops are grown in a two-year interval in CR1 compared to cover crops grown every 4th year in CR2. Among cover crops, winter rye showed the highest C inputs (Supplementary Table S7), with additional 20 and 24 % C input for crop rotation CR1





and 9-11% for crop rotation CR2. These are followed by saia oat (15-17% for CR1 and 7-8% for CR2) and spring vetch (11-12% for CR1 and 5% for CR2).

The influence of cover crops on soil C sequestration was determined from the difference between the respective cover crop
scenario and the control variant for crop rotation CR1 and CR2 (Figure 4, Table S7). As expected, the C sequestration rates are about twice as high if the cover crop is cultivated in a two-year interval (CR1) compared to a four-year interval CR2). Focusing on CR1, annual C sequestration rates averaged for the simulation periods of 50 years were highest for winter rye with 0.13 (0.06; 0.2) t C ha$^{-1}$ a$^{-1}$ and 0.15 (0.1; 0.2) t C ha$^{-1}$ a$^{-1}$ for Göttingen and Hohenheim, respectively, followed by saia oat (0.11 (0.06; 0.2) C ha$^{-1}$ a$^{-1}$ in Göttingen and (0.12 (0.08; 0.2) C ha$^{-1}$ a$^{-1}$ in Hohenheim) and spring vetch with 0.09 (0.04;
0.17) C ha$^{-1}$ a$^{-1}$ for both sites (with minimum and maximum values of the model ensemble in brackets). Differences between sites and treatments are small compared to the simulated variability caused by model approaches contained in the applied multi model ensemble.

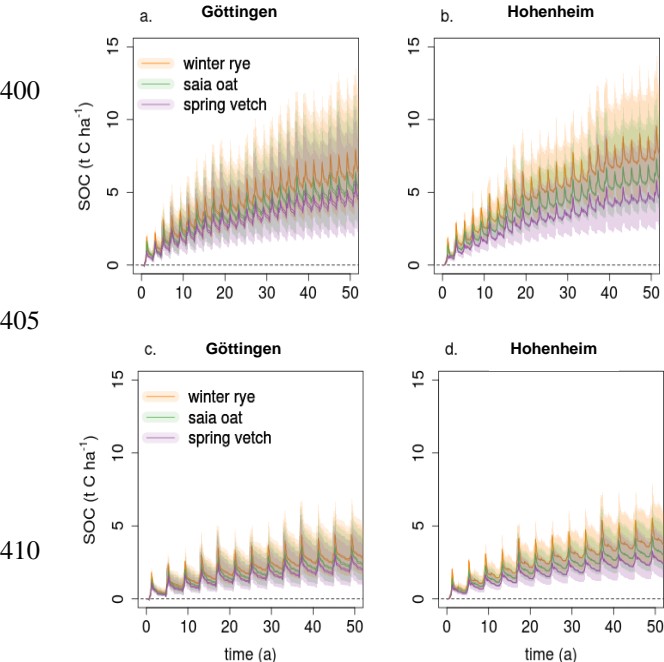

**Figure 4.** Modelled effect of cover crops on the increase in SOC stocks (0-30 cm) for regionally common crop rotations (a, b) CR1 "cover
crop/bare fallow – sugar beet – winter wheat – cover crop/bare fallow – silage maize with an application of 30 m³ digestate from biogas plants before seeding of the maize" and (c, d) CR2 "cover crop/bare fallow – sugar beet - winter wheat – winter rape – winter wheat" with a cover crop in every 4th year at the Göttingen and Hohenheim experimental sites (2 fields per site). Shaded areas show the variability of the model ensemble.






**Table 5**. Simulated carbon sequestration rates caused by cover crops, averaged over a simulation period of 50 years with model structural uncertainties in brackets

| Crop rotation | Cover crop | Carbon sequestration rate by cover crops [t C ha$^{-1}$ a$^{-1}$] | | Carbon sequestration rate by cover crops [kg CO$_2$ ha$^{-1}$ a$^{-1}$] | |
|---|---|---|---|---|---|
| | | Göttingen | Hohenheim | Göttingen | Hohenheim |
| CR1 | Saia oat | 0.11 (0.06;0.2) | 0.12 (0.08;0.2) | 403 (220;733) | 440 (293;733) |
| CR1 | Spring vetch | 0.09 (0.04;0.17) | 0.09 (0.05;0.17) | 330 (147;623) | 330 (183;623) |
| CR1 | Winter rye | 0.13 (0.07;0.22) | 0.15 (0.1;0.24) | 477 (257;807) | 550 (367;733) |
| CR2 | Saia oat | 0.06 (0.03;0.1) | 0.07 (0.05;0.11) | 220 (110;367) | 257 (183;403) |
| CR2 | Spring vetch | 0.05 (0.02;0.09) | 0.06 (0.03;0.1) | 183 (73;330) | 220 (110;367) |
| CR2 | Winter rye | 0.06 (0.04;0.11) | 0.08 (0.05;0.13) | 220 (147;403) | 293 (183;477) |

## 4 Discussion

### 4.1 Effect of cover crops on soil water content

Concerns have been raised about the potential for increased evapotranspiration from cover crops to adversely impact soil moisture (Unger and Vigil, 1998). In our study, particularly in the G18 trial, which received only 430 mm of rainfall in 2018, 195 mm below the long-term average, we observed that mean soil moisture levels, as indicated by WFPS, were lower in plots with cover crops compared to fallow during the cover cropping phase. Although these differences were not statistically significant, they align with observations from arid and semi-arid regions, suggesting that under conditions of annual rainfall less than 500 mm, soil water storage may decrease with the use of cover crops, potentially leading to reductions in the following crop yield (Blanco-Canqui et al., 2015; Mitchell et al., 2015). However, we did not observe any differences in WFPS among treatments in other site-years during the cover cropping phase or in subsequent phases. This observation is supported by the findings of Wang et al. (2021), who concluded that the impact of cover crops on pre-sowing soil moisture, soil water storage potential, and main crop yield varies greatly depending on the specific site conditions and regional climate. Whereas in regions with adequate precipitation, like humid and subhumid areas, cover crops may have minimal effect on soil water storage potential (Qi et al., 2011). This phenomenon could be attributed to an increase in soil water holding capacity by increased soil organic matter under cover crops, which may compensate for the increased evapotranspiration, thereby not affecting overall topsoil moisture levels (Poeplau and Don, 2015). Additionally, Basche et al. (2016) have demonstrated that the long-term incorporation of a rye cover crop can significantly enhance field capacity water content and plant-available water. Similarly, In a parallel study conducted within the same experimental framework, Grunwald et al. (2022) found that winter cover crops, particularly those with deeper and more extensive root systems like rye, improve soil aggregate stability, reduce penetration resistance, and increase soil water content by April. These findings suggest that such cover crops may create favorable conditions for the early growth of subsequent crops such as sugar beet.



## 4.2 Influence of cover crop species on SMN dynamics


In our study, the cultivation of field pea as the preceding main crop led to elevated SMN levels in the autumn, which were substantially higher in Göttingen than in Hohenheim. Subsequently, we observed an increase in SMN across all treatments and site-years for several weeks following the incorporation of pea straw (see Fig. 1). This trend can be attributed to the net mineralization of the pea residues, a common characteristic of legumes with low C:N ratios that promote swift decomposition

and N release (Doran and Smith, 1991).

Consistent with our hypothesis, non-legume cover crop cultivation led to a significant reduction in SMN levels. By late November, we found that SMN levels were considerably lower in plots with cover crops compared to those left fallow. This outcome aligns with previous findings, which demonstrated the efficacy of non-legume cover crops in scavenging SMN, thereby reducing the risk of N leaching during winter months (Helfrich et al., 2024; Nouri et al., 2022).

Among the cover crops evaluated, rye emerged as most effective in reducing SMN across all site-years, a consequence of its higher biomass production and N uptake. Rye's resilience to cold weather, in contrast to the frost-sensitive oat and vetch, enables it to withstand frost events without damage, continuing to accumulate biomass and absorb N even under less favorable conditions. This observation confirms the findings of Thapa et al. (2018), who identified a positive correlation between the biomass of cover crops and their $NO_3^-$ uptake.

The differences in SMN levels between rye and fallow treatments by the end of November were high, reaching approximately 130 and 80 kg N ha$^{-1}$ in the G18 and G19 trials, respectively, and averaging around 30 kg N ha$^{-1}$ in both H18 and H19. The more pronounced differences in the Göttingen trials reflect the higher initial SMN values there compared to Hohenheim. Echoing the insights from this study, Koch et al. (2022) observed that the difference in SMN levels between rye and fallow treatments across the 0-90 cm soil profile in H18 closely matched the N content found in rye shoots. Interestingly, for H19,

the N content in rye shoots exceeded the difference in SMN levels between the rye and fallow treatments, highlighting the efficiency of rye to uptake N, which likely contributed to mitigating $NO_3^-$ leaching. Conversely, in the Göttingen trials (G18 and G19), the N content in rye shoots accounted for merely half of the difference in SMN levels between rye and fallow treatments. Our study did not account for the root biomass, root exudates and N immobilization, which might explain some of the observed reduction in SMN levels under cover crop treatments. This variation between the two sites could result from the

higher initial SMN content in Göttingen and the potential limitations on the overall N uptake capacity of the cover crops.

In contrast, vetch emerged as the least effective cover crop in reducing SMN levels, a distinction that can be traced back to its shallower root system, especially when compared to oat and rye (Grunwald et al. 2022). This characteristic, coupled with the capacity of vetch for biological N fixation likely contributed to its reduced N uptake (Ramirez-Garcia et al. 2015). Moreover, winter legume cover crops tend to produce less biomass than legumes grown in summer, exerting a diminished impact on soil

N uptake as also documented earlier (Pan et al. 2022).

From November through January, we observed a consistent decline in SMN levels across all treatments in every trial, reaching their lowest level during this period. This trend was most pronounced in plots designated as fallow, where SMN levels in





November were the highest. The phenomenon of $NO_3^-$ leaching beyond the measurement depth could explain a portion of the observed decrease in SMN during the winter months. However, since our study refers solely to the topsoil layer, it is

challenging to fully assess $NO_3^-$ leaching from the entire root zone. Supporting this observation, Nouri et al. (2022) demonstrated that cover crops, on average, reduce $NO_3^-$ leaching by 69% in comparison to fallow conditions, highlighting the significant role of cover crops in enhancing N retention within the soil profile.

Following January, as anticipated in our hypothesis, SMN levels began to rise, with the increase being more pronounced in treatments involving frost-sensitive oat and vetch. This rise in SMN may be attributed to the breakdown of organic matter

caused by frost damage, which leads to the physical deterioration of plant material and accelerates microbial decomposition, especially in frost-sensitive cover crops. Additionally, enhanced soil N mineralization driven by increased soil moisture during this period likely contributed to the observed increase in SMN. Consequently, in February, SMN levels in the oat treatment were significantly higher than those observed in the fallow and in the frost-resistant rye treatments in three out of four trials. These observations are consistent with findings by Storr et al. (2021), who reported an increase in SMN during winter,

correlating with a reduction in N stored in the shoot biomass of cover crops following frost-induced mortality and senescence in frost-sensitive species.

From February to April, a decline in SMN was recorded across all treatments, coinciding with a period of high precipitations and elevated soil moisture, which may suggest an increased risk for $NO_3^-$ leaching into deeper soil layers. By the end of the cover cropping phase in April, the lowest SMN values were recorded in plots with the rye treatment, although the variances

did not reach statistical significance in most site-years.

Upon evaluating the data across all site-years for the cover cropping phase, we observed a reduction in average SMN for the non-legume cover crops specifically rye and oat, whereas vetch did not exhibit a similar impact. This outcome aligns with previous research indicating lower SMN uptake by legumes compared to non-legumes, a difference that may be partially attributed to the capability of legumes to fix atmospheric N, supplementing their absorption from the SMN pool (Daryanto et

al., 2018; Helfrich et al., 2024; Ramirez-Garcia et al., 2015).

The application of N fertilizer to the sugar beet following the incorporation of the cover crops into the soil strongly influenced the SMN content in the topsoil, making the effect of the cover crops no longer detectable. During the sugar beet phase and the subsequent winter wheat phase, we did not detect any significant differences in SMN among the treatments. This observation suggests that on an annual scale, within a fertilized agricultural framework, the presence of winter cover crops may not

substantially influence N availability for subsequent cash crops after fertilization. On the other hand, in an unfertilized setting, cover crops exert a pronounced impact on SMN during the sugar beet phase, thereby affecting yield (Koch et al., 2022; Kühling et al., 2023).

## 4.3 Influence of cover crop species on N₂O flux rates

Among the crucial factors influencing $N_2O$ production and emissions in agricultural soils are the availability of SMN in the

forms of $NO_3^-$ and $NH_4^+$, which serve as substrates for nitrification and denitrification processes, as well as suitable soil





moisture and temperatures conducive to microbial activity. Despite the initial weeks of the trials showing elevated levels of SMN and the presence of adequately warm soil temperatures, our study did not observe high $N_2O$ flux rates. This discrepancy may be attributed to the low levels of WFPS recorded during the late summer period. In line with this finding, Cosentino et al. (2013) have suggested that WFPS values below 59% serve as a limiting factor for $N_2O$ production, and Smith et al. (2003)

have delineated an increase in $N_2O$ flux rate with increasing WFPS, providing that SMN content was not a limiting factor. Throughout the cover cropping phase, $N_2O$ flux rates remained relatively low. This is attributed to the suboptimal conditions for microbial activity prevalent during the autumn and winter months, such as reduced temperatures and the limited availability of C and N. Cosentino et al. (2013) emphasized the critical role of topsoil temperature in $N_2O$ production, indicating that emissions were relatively low when soil temperatures declined below 14°C, irrespective of other soil characteristics.

Furthermore, Rummel et al. (2021) demonstrated that soil moisture does not increase $N_2O$ emissions when $NO_3^-$ is limited. An important benefit of frost-sensitive cover crops cultivation is their natural termination under appropriate winter conditions, eliminating the need for chemical termination through herbicides. However, the stage at which cover crops are terminated has a significant impact on the characteristics of plant residues, including their rate of mineralization and, consequently, the potential for promoting $N_2O$ emission. Abalos et al. (2022) highlighted the influence of residue maturity stage at termination

on $N_2O$ emissions, with immature residues leading to higher emissions. In our research, as hypothesized, we observed increased $N_2O$ flux rates during the winter months, when frost-sensitive cover crops were terminated by frost. A frost event at the end of January 2019, which resulted in the termination of oat and vetch, was followed by pronounced $N_2O$ flux rates from the oat treatments in both G18 and H18 trials. Similar increases in $N_2O$ flux rates were recorded for vetch in G18 but not in H18, with the severity and duration of the frost event being more pronounced in G18, leading to greater $N_2O$ emission peaks. This increase

in cumulative $N_2O$ emissions from frost-sensitive cover crops during winter can be attributed to the degradation of organic matter as a result of physical damage by frost, followed by microbial decomposition. This process not only increases N availability but also promotes microbial respiration, potentially leading to the formation of anoxic microsites conducive to enhanced $N_2O$ production (Beauchamp et al. 1989; Chen et al. 2013; Kravchenko et al., 2017).These observations align well with the work of Wagner-Riddle et al. (2017), who found that cumulative annual $N_2O$ emissions in cold regions were closely

linked to the number of freezing-degree days, highlighting the impact of soil temperature dynamics on $N_2O$ emissions. Numerous studies have documented a swift increase in $N_2O$ emissions following N fertilizer application (Dobbie and Smith, 2003; Weitz et al., 2001). The addition of organic amendments aimed at enhancing soil fertility and crop productivity can trigger greenhouse gas emissions through mechanisms like the priming effect, nitrification and denitrification (Thangarajan et al., 2013). Abalos et al. (2022) highlighted that, on average, the incorporation of cover crop residues results in a 50% increase

in $N_2O$ emissions. These findings are in accordance with those of Mutegi et al. (2010), who reported that 60% of annual $N_2O$ emissions were a consequence of tillage and residue incorporation. In our investigation, following the incorporation of cover crop residues into the soil, coupled with the sowing and N fertilization of sugar beet, we observed an increase in $N_2O$ flux rates spanning six to eight weeks across all treatments. Despite similar elevated SMN levels across treatments due to N fertilization, $N_2O$ flux rates were significantly higher following the rye treatment compared to other treatments. Conversely, $N_2O$ flux rates



from the oat and vetch treatments were considerably lower and on par with those observed in the fallow treatment, displaying their potential in reducing $N_2O$ emissions during this phase. Unlike the shallow tilling to 15 cm applied in other treatments, rye required deep ploughing up to 30 cm to manage its extensive root system. The effect of tillage intensity on $N_2O$ emissions may depend on soil and climatic conditions. In some areas, reduced tillage promotes $N_2O$ emissions, while in others it may reduce emissions or have no measurable effect (Marland et al., 2001). While Boeckx et al. (2011) suggested that tillage

intensity does not significantly impact $N_2O$ emissions, Mutegi et al. (2010) reported that conventional tillage systems produced higher $N_2O$ emissions than reduced tillage or no-tillage systems, presumably due to increased soil porosity and gas diffusivity under conventional practices that might enhance $N_2O$ release to the atmosphere. On the other hand, Rochette et al. (2008) reported that ploughing heavy clay soils could substantially reduce $N_2O$ emissions, likely due to enhanced soil porosity, which supports better soil aeration and moisture levels, thereby limiting denitrification and $N_2O$ production. However, in a study by

Grunwald et al. (2022) examining the impact of cover crops on soil structure within the Göttingen trials (G18 and G19), it was observed that ploughing after rye, in comparison to cultivator tillage following other cover crops, led to only a slight and statistically insignificant increase in soil porosity across most cover crop treatments. This suggests that variations in the incorporated cover crop biomass are likely the primary drivers behind the differences in $N_2O$ emissions subsequent to incorporation. The larger biomass of incorporated fresh rye residues, in comparison to other treatments, is likely to have

resulted in increased carbon turnover and heightened microbial activity. This, in turn, resulted in a more rapid depletion of soil oxygen, creating anaerobic microsites that favored higher $N_2O$ emissions (Blagodatsky et al., 2011). Water-filled pore space exerted a notable influence on $N_2O$ flux rates throughout the sugar beet phase. During the sugar beet phase, heavy rainfall increased $N_2O$ flux rates. However, additional N fertilization in the rye treatment in June for the G19 trial and across all treatments in the H18 trial raised SMN levels, but $N_2O$ flux rates were significantly lower compared to earlier in the season

when WFPS was higher. From July to the conclusion of the sugar beet phase, $N_2O$ flux rates nearly diminished to non-detectable levels, likely due to a combination of reduced WFPS and minimal SMN values, a reflection of the robust N uptake by the maturing sugar beet plants.

## 4.4 Cumulative $N_2O$ emissions

Despite non-legume cover crops significantly reducing SMN levels compared to bare fallow during the cover cropping phase,

their cumulative $N_2O$ emissions did not decrease accordingly. Contrarily, the frost-sensitive oat exhibited significantly higher cumulative $N_2O$ emissions than fallow during this phase. Our analysis indicates that SMN levels did not significantly influence $N_2O$ emissions during the cover cropping phase, suggesting that SMN concentration alone do not govern $N_2O$ emissions. Instead, a combination of factors including temperature, WFPS, and the availability of C and N plays a critical role in modulating $N_2O$ emissions. The relatively low $N_2O$ emissions observed during the cover crop phase could be attributed to

unfavorable conditions for microbial activity, such as low temperatures and a limited availability of readily decomposable C and N sources.





In our comparison of cumulative $N_2O$ emissions from frost-sensitive cover crops, we found that our initial hypothesis, that residues with lower C:N ratios cause higher $N_2O$ emissions, was not supported. Instead, the C and N contents, which correlated with the dry matter of the residues, emerged as a more reliable indicator in this study. Despite oat having significantly higher

C:N ratios compared to vetch, it still induced higher cumulative $N_2O$ emissions when its C and N contents were higher. Despite the differences in C:N ratios between oat and vetch, both fell within a range known to facilitate net mineralization and increase soil $NO_3^-$ content, thereby promoting $N_2O$ losses (Li et al., 2013). This observation aligns with the findings of Millar and Baggs (2004) and Li et al. (2013), indicating that a greater release of readily available C and N from residues with similar C:N ratios results in increased microbial activity and consequently, higher $N_2O$ emissions.

During milder winters, such as observed in the H19 trial, cover crops persisted through the season, leading to no notable differences in cumulative $N_2O$ emissions among the treatments during the cover cropping phase. This suggests that without early frost termination, frost-sensitive cover crops share comparable effects on $N_2O$ emissions with their frost-resistant counterparts. This observation aligns with the findings of Storr et al. (2021), who determined that frost-sensitive cover crop species may not always terminate under temperate climates, but they continue to provide a steady supply of available C and N

as the plants senesce.

Wagner-Riddle et al. (2017) estimated that thaw emissions could account for 35-65% of the total annual $N_2O$ emissions from seasonally frozen croplands. In this study, frost-induced $N_2O$ emissions were observed in plots with frost-sensitive cover crops but not in fallow or plots planted with frost-resistant rye. This pattern suggests that the $N_2O$ emissions in this context are primarily driven by the mineralization of frost-sensitive cover crop residues and subsequent increased microbial activity within

the soil. However, the short-lived nature of these thaw-induced $N_2O$ fluxes, combined with the lack of daily measurement intervals, potentially led to an underestimation of cumulative winter $N_2O$ emissions. The employment of automatic continuous-flow chambers, capable of sampling several times daily, could have yielded a more accurate estimation of these emissions.

As postulated, rye, with the largest biomass among the cover crops, was associated with the highest cumulative $N_2O$ emissions following the incorporation of its residues, despite exhibiting the highest C:N ratios. According to the IPCC emissions factor

for wet climates, approximately 0.6% of the N present in crop residues is expected to be converted into $N_2O$ (IPCC, 2019). In the case of rye, with an average shoot biomass N content of 103 kg N ha$^{-1}$, 0.6% thereof would account for only 21% of the 2.9 kg $N_2O$-N ha$^{-1}$ increase compared to the fallow treatment during the sugar beet phase. Rye root decomposition as well as enhanced soil decomposition could account for part of the surplus $N_2O$ emissions. Li et al. (2015) observed that residues with high C:N ratios could still lead to significant $N_2O$ emissions if the C present is easily decomposable. Abalos et al. (2022)

identified N and easily degradable organic fractions as key factors affecting $N_2O$ emissions from crop residues. Therefore, the elevated cumulative $N_2O$ emissions observed following the incorporation of rye can be attributed to the high biomass yield and the large amounts of readily decomposable carbon and nitrogen.

It is noteworthy that the cumulative $N_2O$ emissions during the winter wheat phase were significantly lower in the rye treatment in comparison to the fallow and vetch treatments, which exhibited higher levels of cumulative emissions.



Even though SMN levels were generally higher in Göttingen, the cumulative $N_2O$ emissions tended to be higher in the Hohenheim trials (H18, H19). This difference might be attributed to the higher clay content in Hohenheim soils, which likely reduced gas diffusivity, resulting in more anaerobic microsites and, therefore, elevated denitrification rates, a factor known to increase $N_2O$ emissions (Bollmann and Conrad, 1998; Pelster et al., 2012).

Throughout the trial period, and due to the sugar beet phase, rye cover crop was associated with higher cumulative $N_2O$
emissions, although the increase was not statistically significant. These results are consistent with those of Basche et al. (2014), who observed that cover crops have a net neutral impact on $N_2O$ emissions when measured over longer timescales.

### 4.5 Climate mitigation potential of winter cover crops

### 4.5.1 Mitigation potential of indirect $N_2O$ emissions by non-legume cover crops

Cover crops, particularly during periods of high soil N availability and high precipitation, have the potential to absorb excess
N and reduce the risk of $NO_3^-$ leaching, thus mitigate indirect $N_2O$ emissions. While cover crop cultivation in this study did not diminish direct $N_2O$ emissions and, in the case of rye, slightly intensified them, it potentially led to a reduction in indirect $N_2O$ emissions by reducing $NO_3^-$ leaching. The potential mitigation of indirect $N_2O$ emissions during the cultivation of non-legume winter cover crops in our study was on average 0.27 kg $N_2O$-N ha$^{-1}$ a$^{-1}$ for rye and 0.2 kg $N_2O$-N ha$^{-1}$ a$^{-1}$ for oat, equivalent to 116 and 85 kg $CO_2$ ha$^{-1}$ a$^{-1}$, respectively. This finding is in line with Parkin et al. (Parkin et al., 2016) who noted
that rye, while being neutral regarding direct $N_2O$ emissions, substantially decreased indirect $N_2O$ emissions over a decade-long trial.

The methodology employed in the present study, which utilizes nitrogen uptake by winter cover crops, offers a means of estimating the potential for indirect $N_2O$ emissions to be mitigated. However, the model does not take into account soil water movement or nitrogen concentration in leachate. Nevertheless, it provides an approximate estimation of the potential mitigation
of indirect $N_2O$ emissions through the cultivation of cover crops. Additionally, our analysis solely considered the N present in the aboveground biomass of the cover crops, disregarding the N stored in the roots or the N remineralization occurring during the winter period. Furthermore, the level of mitigation varies based on the soil type and climatic factors, with sandy soils and high precipitation periods exhibiting more pronounced effects (Simmelsgaard, 1998).

### 4.5.2 Effect of cover crops on long-term soil C sequestration

According to our model results, soil C sequestration from winter rye, saia oat and spring vetch significantly contributes to greenhouse gas mitigation by cover crops and is a relevant sink compared to direct and indirect $N_2O$ fluxes. Sequestration rates found here for a profile depth of 30 cm are in the same order or a bit smaller than sequestration rates reported in several meta-studies (Abdalla et al., 2019; Blanco-Canqui, 2022; Bolinder et al., 2020; Poeplau and Don, 2015). In the meta-study by Poeplau and Don (2015), a sequestration rate of 0.32±0.08 Mg C ha$^{-1}$ a$^{-1}$ was found in the 0-22 cm profile depth using a linear
regression approach. The 37 experiments analyzed in Poeplau and Don (2015) represented tropic and temperate regions and



cover crops were grown annually. Assuming proportionality between C input and soil C stock changes and a uniform SOC distribution in the plough horizon, this would translate into $0.21\pm0.054$ Mg C ha$^{-1}$ a$^{-1}$ and $0.11\pm0.04$ Mg C ha$^{-1}$ a$^{-1}$ for a cover crop every second and fourth year. Averaged sequestration rates calculated for the fields experiments analyzed in this study vary between 0.09 - 0.15 Mg C ha$^{-1}$ a$^{-1}$ and 0.05 - 0.08 Mg C ha$^{-1}$ a$^{-1}$, which is lower compared to the regression approach from
Poeplau and Don (2015) but this is consistent with lower C inputs from cover crops in this study (CR1: 0.82 Mg ha$^{-1}$ a$^{-1}$ -1.82 Mg ha$^{-1}$ a$^{-1}$) compared to reported mean C inputs from cover crops of 1.87 (95% confidence interval: 1.65 – 2.09) Mg ha$^{-1}$ a$^{-1}$ in Poeplau and Don (2015). One probable reason for the low C inputs from cover crops in Göttingen and Hohenheim is the fact, that cover crops were sown relatively late (end of August - begin of September) which hampered optimal biomass production.

The models used are pure soil C models, which lead to a number of restrictions and uncertainties. Plant growth as an output variable of the C input calculation is not modelled but is estimated using experimental variables and values from statistics. For C input modelling from cover crops in particular, there is uncertainty as to whether the required average cover crop biomass is well represented by two annual values from the rather dry years 2018 and 2019. In addition, N availability may play a role in biomass growth and C utilization efficiency in the conversion of crop residues to soil organic matter (Jian et al., 2020). This
is not taken into account by the models used.

These considerations are crucial in the selection of cover crops, highlighting the intricate relationship between N dynamics, residue decomposition, and microbial activities. This complexity emphasizes the need for a comprehensive understanding of cover crop selection and management to optimize N uptake efficiency and reduce N$_2$O emissions. Modeling approaches could serve as a valuable tool in this context, offering predictions on the most suitable cover crop types based on specific site
conditions such as soil type, climate, and the type of main crop that follows the cover crops. Conducting incubation studies under controlled conditions with labelled N can further our knowledge of the primary factors driving N$_2$O emissions, facilitating more informed decisions regarding cover crop selection tailored to local conditions. Given these insights, achieving a balance between maximizing N capture during the cover cropping phase and minimizing N$_2$O emissions during residue incorporation becomes paramount. There is a compelling need for additional research into the mechanisms behind higher N$_2$O
emissions from cover crop residues and for developing strategies to mitigate these effects. Investigating the influences on residue decomposition rates, microbial activity, and N cycling processes will offer deeper understanding of the patterns of emissions observed, leading to the advancement of more sustainable agricultural practices.

## 5 Conclusions

Our study highlights the complex role of cover crops in agricultural systems, particularly in relation to soil N dynamics, N$_2$O
emissions and C sequestration. While cover crops, especially non-legumes like rye and oat, have demonstrated significant potential in reducing SMN levels and mitigating the risk of NO$_3^-$ leaching, their impact on N$_2$O emissions is multifaceted. Our findings highlight that frost-sensitive cover crops can lead to increased N$_2$O emissions following frost events or, in all cases,



the incorporation of crop residues for establishing the next cash crop combined with mineral N fertilization. However, the potential of cover crops to mitigate indirect $N_2O$ emissions and sequester C, suggests a beneficial aspect of their use in

sustainable agriculture. This balance between N capture, $N_2O$ emission and C sequestration, emphasizes the need for strategic management of cover crops to harness their benefits fully while minimizing potential environmental drawbacks. Especially, the type of cover crop (frost tolerance, legume or non-legume and pest control aspects) needs to be chosen carefully depending on the following cash crops and climatic conditions. Future research should focus on developing crop rotation and site-specific management practices that optimize cover crop benefits for soil health, crop productivity, and climate change mitigation.

**Author contributions.** VN conceived of the research, contributed to methodology development, performed data curation, performed formal calculations, data analysis and investigation, and drafted the original manuscript. MH provided critical input during the review and editing process, contributed resources and validation, and participated in drafting the manuscript. RD performed formal analysis, provided visualization support, and participated in drafting the manuscript. AM and PR contributed to the review and editing of the manuscript. H-JK provided supervision throughout the project, contributed to the review and
editing process, and played a role in the design of the study. RR provided feedback, supervised the project at the Hohenheim site, contributed to the conceptualization, and acquired partial funding for the research, while LE performed data collection and curation at the Hohenheim site. KD provided feedback, supervised the project at the Göttingen site, contributed to the conceptualization, and acquired partial funding for the research. All authors reviewed and approved the final version of the manuscript.

**Competing interests.** The authors declare that they have no known competing financial interests or personal relationships that could have appeared to influence the work reported in this paper.

**Acknowledgements.** This study was conducted as part of the joint research project "Reduction of greenhouse gas emissions from crop production through site-specific optimized cover crop systems" (THG ZwiFru), funded by the German Federal Office for Agriculture and Food on behalf of the Federal Ministry of Food and Agriculture (Funding reference no.
281B200716). We extend our sincere appreciation to the local farmers in Göttingen and the crew at the Institute for sugar beet research in Göttingen for facilitating our experiments on their fields. Special thanks go to Thorsten Gronemann and Marlies Niebuhr for their assistance in the field, and to Dennis Grunwald for his efforts in data curation

**Financial support.** This project has been funded by the Federal Ministry of Food and Agriculture (BMEL), following a decision by the Parliament of the Federal Republic of Germany, and is administered through the Federal Office for Agriculture
and Food (BLE) under the innovation support program (Funding reference no. 281B200716). This work was partially funded by the Deutsche Forschungsgemeinschaft (DFG, German Research Foundation) under Germany's Excellence Strategy – EXC 2070 – 390732324.

**Data availability.** Data are available here: https://doi.org/10.25625/HFEDA7

**Supplement.** The supplement related to this article is available online at doi…



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
