# Peer review of "Evaluating N2O Emissions and Carbon Sequestration in Temperate Croplands with Cover Crops: Insights from Field Trials"

_EGUsphere, 2024_

## Referee Comment (RC1)

With pleasure I read the manuscript entitled 'Managing Soil Nitrogen Surplus: The Role of Winter Cover Crops in $N_2O$ Emissions and Carbon Sequestration'. There are still quite some knowledge gaps to fill regarding this topic and the authors contributed to bridging some of these gaps. In general, I think the manuscript is publishable after some minor revisions. Especially the carbon modelling needs more explanation. More specific comments are listed below.

L26-30: the field trial was only 16 months, but you estimated carbon sequestration over a 50 year time period. I guess the authors used simulation models to assess the potential long-term sequestration. But this should be added to the abstract.

L31-33: the authors recommend "optimized cover crop selection", but according to the results, not much difference in N2O emission and C sequestration is noticed between the different cover crop varieties.

L75-77: this is a stand-alone statement which comes out of the blue. Elaborate on it (because the authors also focused on the short-term N2O emissions), or delete the sentence (or move it to the discussion).

L81: "soil organic models", I think the word 'carbon turnover' or 'matter turnover' is missing here.

L109: I'd recommend to describe the soil characteristics for each Luvisol separately, so a reader knows which soil has a soil organic matter content of 20 g/kg and 30g/kg.

L115-135: it would help if the experimental design is accompanied with a table.

L154: please add a reference to your assumption of 2.65 g/cm3.

L156: please explain why you used two types of chambers. Can the results still be compared, because the volume of the two chambers differs?

L159: add 'N2O fluxes' between 'measured and 'using dark'. Again, how can you compare these results with the results from the other chambers used in Gottingen?

L172: replace (IPCC, 2019) for IPCC (2019)

L201: How do you know the effect in C stock change is caused by the addition of cover crops when you also apply other organic inputs (30m3 digestate) at maize?

L206: I miss some information on the modelling using RothC and C-Tool. At the moment I'm not able to reproduce your modelling exercise. How did you use both tools (e.g., in an ensemble run or did one model complement the other)? How did you initialize the SOC stock? What historical management took place on the fields? Which site-specific input data did you require/use and which input data did you assume (e.g., soil depth, climate data, cover factor (what assumption did you make)? Is there any irrigation in the fields, or ploughing? What are the soil properties? Why did you choose for this model and not for a model that assesses C and N fluxes?

L217: the source you refer to studied tree species. How applicable is this approach for green manure and more specifically to the green manure types that were included in this study?

L219: replace 'a parameter' for 'a plant-specific parameter'

L226: why did you decide to copy the weather data 2018-2021. These were extremely dry years and might not be representative on the long term (as also mentioned in L230). Consider climate scenario's or a longer time range.

L250: it is not clear to me how the site-specific weather data differ from the DWD weather data. Also explain in Chapter 2.5 why you used DWD instead of site-specific weather data. I agree with your decision, but it might cause some confusion.

L285: the author did not mention the N fertilization of sugar beets before. This should be added to the methodology.

L321: why did vetch show N2O peaks, and why did only G18 show peaks and G19 not?

L415: the text below and the figures do not match. I'd expect two scenario's, one for CR1 and one for CR2, and the baselines (controls). Please, clarify the modelling approach.

L430: linking the results to research done in a completely different climatic zone requires more explanation or needs to be removed.

L445: in Chapter 4.2 some results are mentioned. Consider combining the Results and Discussion section or move the results to Chapter 3.

L622-624: do not repeat the results

L635 – 644: do not repeat the results. Re-write this section and try to be more concise.

Due to the high number of hypotheses, the Discussion is exhaustive and good, but extremely long. Perhaps consider a restructuring and start with an overview of the hypotheses (rejected or accepted) followed by a discussion and underpinning of the results for each hypothesis.

---

## Author Response (AR1)

10th March 2025

Dear Editors and Reviewers,

Thank you for your valuable feedback on our manuscript. In the revised version, we have implemented the following changes as detailed in our "Reply on RC1 & Reply on RC2" (Victoria Nasser, 21 Jan 2025):

- We implemented all revisions suggested by reviewers 1, 2, and the topic editor.
- We combined the separate Results and Discussion sections into one integrated "Results and Discussion" section.
- Hypotheses are now referenced with numbers in the discussion.
- We edited the text of the "Results and Discussion" to be more concise by removing redundant descriptive sentences and avoiding iterations, which shortened the Results and Discussion section from approximately 6500 to 4150 words.
- The standalone "3.5 Synthesis over all site-years" section was removed and its content was integrated into sections 3.1 and 3.2 of the revised manuscript. Correspondingly, Table 3 from that section has been moved to the Supplementary Materials as Supplementary Table S5.
- Two new tables were added, as suggested by the reviewers: Table 1, detailing topsoil properties and site information for the different experimental trials, and Table 2, which provides management dates and N fertilizer rates for CC and main crop management across the experimental trials. (Please note that the numbering of tables in both the manuscript and the supplementary materials has been updated accordingly.)
- Several typographical errors have been corrected.
- We have abbreviated "cover crops" as "CCs" throughout the manuscript, following common usage in the literature.
- We ensured that all abbreviations (e.g., nitrogen (N), carbon (C), cover crop (CC), soil organic carbon (SOC)) are used consistently.
- Importantly, we have revised the Materials and Methods sections to consistently employ past tense throughout.
- Additionally, in response to Reviewer #2's comment regarding the title, we have changed the title to more clearly reflect the geographical and methodological scope of our work. This revision aims to prevent any misunderstanding that the paper is a comprehensive review of cover crops across multiple locations and practices.

We believe these revisions have substantially improved the clarity, conciseness, and overall quality of the manuscript. We appreciate your consideration and look forward to your further feedback.

Sincerely,

Victoria Nasser
Georg August University of Göttingen

---

## Author Response (AR2)

**2nd April 2025**

**Author Response to Technical Corrections**

We sincerely thank the executive editor, Dr. Jeanette Whitaker, and the topical editor, Dr. Mart Ros, for their positive evaluation and recommendation to publish our manuscript. We are grateful for the constructive feedback and support throughout the review process.

In response to the final technical corrections:

- Line 77: "depend" has been corrected to "depends."
- Line 89: The word "and" after "hypothesis ii." has been removed.
- Line 107: A space has been inserted between "N" and "content."
- Line 156: The comma after "Flessa et al." has been removed.
- Table 4 and Table 5: Units have been placed in square brackets for consistency with other tables.

We believe these final edits address all outstanding issues, and we thank you again for your consideration and support of our work.

On behalf of all co-authors,
Victoria Nasser